# OpenReview forum: "Indirect Prompt Injections: Are Firewalls All You Need, or Stronger Benchmarks?"
_ICLR.cc/2026/Conference — Submitted to ICLR 2026_

### Official Review · Reviewer_eTDx · 2025-10-28

**Soundness:** 2
**Presentation:** 3
**Contribution:** 1
**Rating:** 2
**Confidence:** 4

**Summary:**

This paper proposes a firewall-based defense method against indirect prompt injection attacks. Specifically, the authors design two LLM-based firewalls: (1) a Tool-Input Firewall that minimizes unnecessary confidential information from tool input arguments, and (2) a Tool-Output Firewall that sanitizes returned tool observations from potentially malicious content. Both firewalls are implemented using LLMs with specifically designed prompts.

The authors evaluate their method on AgentDojo, Agent Security Bench, InjecAgent, and Tau-Bench. The results show that the approach effectively reduces attack success rates while preserving benign utility to some extent. Finally, the authors analyze the strengths and weaknesses of existing prompt injection benchmarks and propose suggestions for improvement.

**Strengths:**

1. The method is simple and effective, requiring no additional training to achieve defense capabilities.
2. The evaluation is comprehensive, covering commonly used datasets in the field.

**Weaknesses:**

1. The novelty is limited. The idea of using LLMs themselves to filter harmful information has been extensively explored in both jailbreak defenses and prompt injection defenses. The authors should clearly articulate what distinguishes their approach from prior work beyond the specific application to tool-input and tool-output filtering.
2. The method causes a degradation in benign utility. As shown in Table 1, benign utility drops dramatically from 83.02% without defense to only 58.41% with the firewall. This suggests the firewall is probably conservative and filters out valuable information necessary for legitimate task completion. The authors should provide a detailed analysis of what types of benign tasks are being incorrectly blocked and whether this trade-off is acceptable in real-world deployments.
3. The paper lacks important analysis on practical deployment considerations: (1) Computational overhead: Since both firewalls are LLM-based, what is the additional latency and cost introduced? This could be prohibitive for real-time applications.
(2) False positive/negative analysis: As mentioned in 2, the paper should provide detailed statistics on misclassification rates to better understand the firewall's reliability.

**Questions:**

Please refer to the weaknesses part.

---

> ### Author Response · Authors · 2025-11-20
> **Response to Reviewer eTDx (1/3)**
>
> We thank the reviewer for recognizing that firewalls are a simple and effective approach that doesn’t require any re-training and appreciating the comprehensive evaluation of benchmarks.
>
> >**W1:** The novelty is limited. The idea of using LLMs themselves to filter harmful information has been extensively explored in both jailbreak defenses and prompt injection defenses. The authors should clearly articulate what distinguishes their approach from prior work beyond the specific application to tool-input and tool-output filtering.
>
> **A1:** We agree that LLM-as-a-judge has been studied in jailbreaking and prompt injection contexts. However, nowhere in the prior work did we find concrete numbers for the firewall baseline on the standard tool-calling security benchmarks (AgentDojo [1], InjecAgent [2], ASB [3]). Results don't necessarily transfer from the Jailbreaking to Indirect Prompt Injection setting: for instance, DoomArena [4] showed that Llamaguard was unable to flag any of the indirect prompt injection attacks they proposed.
>
> **What is a Research Contribution?** Research contributions can take many forms, and are not limited solely to “novel attack strategies” or "novel defense methods". Contributions can also be made through thoughtfully examining existing assumptions in the field (e.g., complex defenses are needed to beat the benchmarks), carefully examining progress thus far (e.g., benchmark limitations), and proposing new directions and considerations that may be needed in the community based on empirical evidence (e.g., more challenging benchmarks). Most importantly, we believe that before proposing new methods, it is equally, if not more important, to extensively and carefully assess the current landscape. Meaningful progress can only be made with a clear understanding of the existing limitations.
>
> **Our Research Contribution is Answering a Scientific Question:** Our work is titled "Are Firewalls All You Need, or Stronger Benchmarks", and it is exactly the scientific question that we try to answer, through extensive empirical analysis on four popular security benchmarks. Our results jointly highlight that (i) a simple firewall-based defense baseline is enough to saturate current security benchmarks, (ii) current benchmarks have several flaws that we fix and provide best practices for future development, and (iii) stronger, more adaptive benchmarks are needed to meaningfully measure progress in agentic security.
>
> **Novel Findings:**
> - **Simple firewalls can easily saturate current benchmarks:** Despite being lightweight, the firewall baseline matches or exceeds the performance of more complex approaches across all 4 major agentic security benchmarks.
> - **The firewalls generalize across models:** We evaluate using four different LLM backbones (GPT-4o, LLaMA 3.3-70B, Qwen3-8B, Qwen3-32B), demonstrating that the firewall-based defense’s effectiveness does not hinge on a specific model or implementation trick.
> - **Current benchmarks have several flaws:** We find and fix the flaws and provide best practices for future development.
>     - For instance, because security benchmarks such as ASB [3] were previously published at ICLR 2025, reviewers refer to it as a strong benchmark for IPI attacks. However, our investigation uncovered several weaknesses and flaws in it (e.g. forcefully injecting an attacker tool in the agent’s planning, lack of arguments), which lead to misconstrued performance and hinder the meaningful measurement of future security progress.
>     - Similarly, the absence of any utility assessment in InjecAgent [2] (accepted at ACL 2024) limits its practical relevance, since an agent that fails to perform its intended tasks could trivially appear perfectly secure.
>     - We also uncover and fix implementation bugs in AgentDojo [1], identify missing or misleading metrics–findings that are new and important for the community.
>
> **Novel Recommendations and avenues for future research:**
> - We suggest that the community use our revised version of AgentDojo instead of the original one (due to minor flaws in evaluation metrics).
> - We ask AI security reviewers to no longer require InjecAgent and ASB numbers, because they may lead to misconstrued performance and hinder the meaningful measurement of future security progress.
> - We urge the community to always report Sanitizer scores as a simple but effective baseline when proposing a new defense method.
> - We urge the community to develop stronger, more adaptive benchmarks, because they are needed to meaningfully measure progress in agentic security.
> - The fact that obfuscation-based strategies can bypass even frontier LLM firewalls (GPT-4o) should be investigated in future work.
>
> **These are all new findings and recommendations that have not been previously published or discussed in the context of tool-calling agent security.**

---

> ### Author Response · Authors · 2025-11-20
> **Response to Reviewer eTDx (2/3)**
>
> >**W2:** The method causes a degradation in benign utility. As shown in Table 1, benign utility drops dramatically from 83.02% without defense to only 58.41% with the firewall. The authors should provide a detailed analysis of what types of benign tasks are being incorrectly blocked and whether this trade-off is acceptable in real-world deployments.
>
> **A2:** We actually recommend using the Sanitizer alone. We don’t recommend using minimizer or combined due to the Minimizer's aggressive redactions. As shown in Table 1, the Sanitizer alone performs significantly better: it preserves high utility (BU: 67.68%, UA: 69.17%) while achieving near-zero ASR (0.02%).
>
> Improved Sanitizer Results: The tradeoff is actually better than initially reported. We developed a new prompt for the Sanitizer, which improves benign utility from 67.68% → 74.09%, closing much of the utility gap while retaining 0% ASR and high utility under attack (69.15%). Based on this, we recommend deploying the Sanitizer alone and not the combined configuration.
> | Defense     | BU (%)           | UA (%)           | ASR (%)           |
> |-------------|------------------|------------------|-------------------|
> | None        | 83.02 ± 5.33     | 50.01 ± 2.59     | 57.69 ± 3.07      |
> | Minimizer   | 70.01 ± 7.76     | 50.03 ± 2.90     | 18.15 ± 1.94      |
> | Sanitizer   | 67.68 ± 3.56     | 69.17 ± 2.47     | 0.02 ± 0.03       |
> | Combined    | 58.41 ± 2.61     | 58.59 ± 1.74     | 0.00 ± 0.00       |
> | **Sanitizer (new prompt)** | **74.09 ± 5.75** | **69.15 ± 2.24** | **0.00 ± 0.00** |
>
> We performed a qualitative analysis of failure cases and found that the Minimizer and Sanitizer sometimes redact information they judge as unnecessary for the current step, but which later turns out to be important for successfully completing the task. We include concrete examples of these problematic redactions for both components in the Appendix G.5. For the Sanitizer, this issue is less pronounced when prompt injection is present in the tool output, as it then focuses primarily on removing the injected content rather than benign task-relevant information.
>
> >**W3:** The paper lacks important analysis on practical deployment considerations: (1) Computational overhead: Since both firewalls are LLM-based, what is the additional latency and cost introduced? This could be prohibitive for real-time applications. (2) False positive/negative analysis: As mentioned in 2, the paper should provide detailed statistics on misclassification rates to better understand the firewall's reliability.
>
> **A3:** We appreciate the suggestion. We performed 3 additional quantitative analyses and 2 additional qualitative analyses:
>
> 1. **Latency analysis:** We present the time required by different LLM backbones to complete the AgentDojo evaluation.
>
> | Model              | Defense    | No attack (s) | With attack (s) |
> |--------------------|-----------|---------------|-----------------|
> | gpt-4o-2024-08-06  | none      | 510.74        | 5917.86         |
> | gpt-4o-2024-08-06  | minimizer | 1257.10       | 13277.98        |
> | gpt-4o-2024-08-06  | firewall  | 1697.19       | 20811.03        |
> | gpt-4o-2024-08-06  | combined  | 2384.43       | 25406.55        |
> | Qwen3-32B          | none      | 1280.41       | 11330.01        |
> | Qwen3-32B          | minimizer | 1421.15       | 11501.99        |
> | Qwen3-32B          | firewall  | 2194.99       | 46781.95        |
> | Qwen3-32B          | combined  | 2628.93       | 49893.98        |
>
> Further, we compare in detail with CaMeL since it is a state-of-the-art system-level defense that employs policies to ensure safety on 2 suites of Agentdojo (banking and Slack) using gpt-4o-2024-08-06. CaMeL requires a total of 8,417.25 seconds to solve the tool-knowledge attack runs, whereas our Firewall needs only 3,347.93 seconds. This corresponds to a 2.5x speed-up over CaMeL under the same attack setting.
>
> | Defense    | No attack (mean) | No attack (total) | Tool-knowledge (mean) | Tool-knowledge (total) |
> |-----------|------------------|-------------------|------------------------|-------------------------|
> | CaMeL     | 32.84            | 838.70            | 65.04                  | 8417.25                 |
> | Minimizer | 28.74            | 736.11            | 41.95                  | 4998.47                 |
> | Firewall  | **27.38**        | **699.90**        | **27.67**              | **3347.93**             |
> | Combined  | 46.75            | 1194.82           | 46.38                  | 5640.89                 |
>
> We have added this analysis in Tables 22 and 23.

---

> > ### Author Response · Authors · 2025-11-20
> > **Response to Reviewer eTDx (3/3)**
> >
> > *(continuation of **A3**)*
> >
> > 2. **Cost overhead:** To evaluate the efficiency of different defenses, we measure the number of input and output tokens used per task on AgentDojo, using gpt-4o-2024-08-06 as the backbone model. We tokenize all messages using OpenAI’s tiktoken tokenizer [8]. Since the native tool-calling API does not expose how it tokenizes tool documentation, we exclude those tokens for all methods, previous ones, and ours. We do include all tokens from the firewall system prompts, ensuring their overhead is fully accounted for.
> >
> > We find that CaMeL incurs the highest token cost among all defenses, requiring 2.82x more input tokens and 2.73× more output tokens than the baseline (no defense) for the median task. In contrast, the Sanitizer alone requires only 0.51x the input tokens and 2.67x the output tokens, showing that it achieves similar or better security with far less prompt overhead, especially in terms of input token usage, which contributes significantly to latency and cost. Full statistics are shown in the table below.
> >
> > | Defense        | Type   | Mean tokens | Median tokens | Std. dev. |
> > |----------------|--------|-------------|---------------|-----------|
> > | None           | Input  | 3764        | 2004          | 4826      |
> > | None           | Output | 303         | 271           | 174       |
> > | Spotlighting   | Input  | 3914        | 2053          | 4698      |
> > | Spotlighting   | Output | 295         | 268           | 169       |
> > | Tool Filter    | Input  | 16562       | 14277         | 12804     |
> > | Tool Filter    | Output | 313         | 293           | 152       |
> > | Repeat Prompt  | Input  | 5854        | 2491          | 8176      |
> > | Repeat Prompt  | Output | 387         | 301           | 324       |
> > | CaMeL          | Input  | 13463       | 3643          | 21276     |
> > | CaMeL          | Output | 1843        | 585           | 2830      |
> > | **Sanitizer**      | Input  | 2036        | 1030          | 2618      |
> > | **Sanitizer**      | Output | 1675        | 723           | 2557      |
> >
> > As shown in the above table, our Sanitizer is much more token-efficient than the system-level defense CaMeL. For input tokens, Sanitizer reduces the mean from 13,463 to 2,036 (≈ 6.6x fewer tokens) and the median from 3,643 to 1,030 (≈ 3.5x reduction). Output token usage is of the same order as CaMeL (mean 1,843 vs. 1,675), so overall, the Sanitizer achieves strong security with substantially lower prompt overhead. We have added this analysis in Table 24.
> >
> > 3. **False Positive and Negative Analysis:** We provide the misclassification statistics that are important for assessing firewall reliability. On AgentDojo, we explicitly compute TP/FP/FN/TN for the Sanitizer:
> >
> > | Component      | Recall  | FNR   | FPR   | Precision | Accuracy |
> > |----------------|---------|-------|-------|-----------|----------|
> > | Sanitizer      | 99.53%  | 0.47% | 0.00% | 100.00%   | 99.82%   |
> >
> > As requested, we provide detailed misclassification statistics in the above table. The Sanitizer demonstrates extremely reliable behavior, with zero false positives and near-perfect recall. The Sanitizer shows a very small false negative rate (0.47%), meaning it misses only a handful of attacks, while maintaining 100% precision (i.e., never misclassifying benign responses). These results support our claim that the Tool Output Firewall is both effective and precise. We have added this to Table 25 in the paper.
> >
> > 4. **Failure examples:** We also provide examples where the Minimizer (Appendix G.5.1) and the Sanitizer (Appendix G.5.2) fail and cause a drop in benign utility. The main reasons are:
> > (a) the Minimizer redacts arguments that are necessary to complete the user’s initial task, and
> > (b) the Sanitizer incorrectly flags tool outputs as unsafe due to the presence of essential PII (e.g., an IBAN) required for solving the task.
> >
> >
> > [1] Debenedetti, Edoardo, et al. "Agentdojo: A dynamic environment to evaluate prompt injection attacks and defenses for llm agents." Advances in Neural Information Processing Systems 37 (2024): 82895-82920.
> >
> > [2] Zhang, Hanrong, et al. "Agent security bench (asb): Formalizing and benchmarking attacks and defenses in llm-based agents." ICLR 2025.
> >
> > [3] Zhan, Qiusi, et al. "Injecagent: Benchmarking indirect prompt injections in tool-integrated large language model agents." ACL 2024.
> >
> > [4] Boisvert, Leo, et al. "Doomarena: A framework for testing ai agents against evolving security threats." arXiv preprint COLM 2025.
> >
> > [5] Debenedetti, Edoardo et al. “Defeating Prompt Injections by Design.” ArXiv abs/2503.18813 (2025).
> >
> > [6] Zhu, Kaijie et al. “MELON: Provable Defense Against Indirect Prompt Injection Attacks in AI Agents.” ICML (2025).
> >
> > [7] Melon Reviews ICML 2025. https://openreview.net/forum?id=gt1MmGaKdZ
> >
> > [8] OpenAI 2022. tiktoken: Fast BPE tokeniser for use with OpenAI’s models. url: https://github.com/openai/tiktoken.

---

> > > ### Comment · Reviewer_eTDx · 2025-11-25
> > > **Response to rebuttal**
> > >
> > > Thank you the authors for their efforts made during the rebuttal.
> > >
> > > > However, nowhere in the prior work did we find concrete numerical data for the firewall baseline on the standard tool-calling security benchmarks.
> > >
> > > Please refer to the work conducted by Shi et al. [1]. In their paper, they clearly demonstrated the efficacy of employing an advanced LLM for prompt sanitization.
> > >
> > > [1] Shi, Tianneng, et al. "Promptarmor: Simple yet effective prompt injection defenses." arXiv preprint arXiv:2507.15219 (2025).
> > >
> > > > Sanitizer (new prompt) results …
> > >
> > > I’m still concerned about the reason behind the 8.95% reduction in benign utilities caused by the sanitizer. Given the relatively low reported false positive rate (0.57%), it’s important to determine if the sanitizer is inadvertently removing essential, benign information. I would appreciate it if some analysis were conducted to address this concern.

---

> > > > ### Author Response · Authors · 2025-11-27
> > > > **Response to Reviewer eTDx**
> > > >
> > > > We thank the reviewer for their continued engagement with our work.
> > > >
> > > > > Further analysis on degradation in BU for the sanitizer.
> > > >
> > > > We conducted a qualitative investigation into the utility degradation under benign settings when using the Sanitizer. Across multiple random seeds, we identified 9 specific user tasks where the BU dropped for at least 2 out of 3 seeds.
> > > >
> > > > We categorized the causes of utility failure as follows:
> > > >
> > > > - **Agent mistake:** The Sanitizer correctly retained the necessary information, but the agent failed to complete the task successfully. These were unrelated to any sanitizer behavior.
> > > >
> > > > - **Sanitizer redaction:** The Sanitizer filtered out content that was necessary for the task completion, leading to a failure.
> > > >
> > > > **Table A.** BU degradation analysis
> > > >
> > > > | Category              | Count |
> > > > |-----------------------|:-----:|
> > > > | Agent mistake         | 7 |
> > > > | Sanitizer redaction   | 2 |
> > > > | **Total**             | **9** |
> > > >
> > > > This breakdown indicates that most of the BU drop stems from agent variability, not overly aggressive redaction by the Sanitizer. We will include this analysis in our revised version for clarity.
> > > >
> > > >
> > > > > On the comparison with PromptArmor [1]
> > > >
> > > > We appreciate the reviewer bringing PromptArmor [1] to our attention. We acknowledge that PromptArmor proposes a defense which is close to our tool-output firewall (Sanitizer).
> > > >
> > > > Based on their reported results, we present the comparison in Table B.
> > > >
> > > > **Table B.** Comparison with PromptArmor on gpt-4o
> > > > | Method        | ASR (%)         |         UA (%)                | BU (%) |
> > > > |---------------|--------------|--------------------|---|
> > > > | PromptArmor   | 0.47        | 68.68             | -|
> > > > | **Ours**          | **0 ± 0**   | **69.15 ± 2.24**  | 74.09 ± 5.75 |
> > > >
> > > > However, PromptArmor does not report benign utility (BU) numbers, making it impossible to fully evaluate the security–utility tradeoff (BU vs ASR). Since BU is a key metric for assessing whether a defense unintentionally impairs normal behavior, the lack of benign-setting results limits a direct comparison.
> > > >
> > > > To further distinguish our work from PromptArmor, we would like to reiterate that our paper aims to answer the broader scientific question *“Are firewalls all you need, or are stronger benchmarks required?”* Through extensive experiments across four widely used security benchmarks, we jointly show that:
> > > > - (i) a simple firewall-based defense baseline is enough to saturate current security benchmarks
> > > > - (ii) current benchmarks have several flaws that we fix for widely adopted benchmarks like AgentDojo and ASB
> > > > - (iii) provide best practices for future development and emphasize the need of more adaptive benchmarks to meaningfully measure progress in agentic security.
> > > >
> > > > We believe that contributions (ii) and (iii) are particularly valuable for the community and are necessary for meaningful progress in the field.
> > > >
> > > > Finally, we note that Shi et al. [1] appeared in July 2025, only about 2 months prior to the ICLR submission deadline. Under standard conference policy, work appearing within 4 months of the deadline is considered concurrent. Therefore, the absence of a numerical comparison against this preprint should not be grounds for rejection or for diminishing the novelty of our contribution.
> > > >
> > > > We are, however, happy to include a discussion of this work in our revised version.
> > > >
> > > > We hope that these clarifications address the novelty concern. If we understand correctly, this appears to be the primary reason motivating the current score. We also believe we have addressed the remaining concerns in the earlier parts of the rebuttal. We would greatly appreciate knowing if there are any outstanding issues that would justify maintaining a score of 2.
> > > >
> > > > &nbsp;
> > > >
> > > > [1] Shi, Tianneng, et al. "Promptarmor: Simple yet effective prompt injection defenses." arXiv preprint arXiv:2507.15219 (2025).

---

### Official Review · Reviewer_MDkm · 2025-11-01

**Soundness:** 2
**Presentation:** 2
**Contribution:** 2
**Rating:** 2
**Confidence:** 4

**Summary:**

The paper proposes a simple defense against indirect prompt injection (IPI) attacks in tool-using LLM agents. It introduces two lightweight modules — a Tool-Input Firewall (Minimizer) and a Tool-Output Firewall (Sanitizer) — that filter inputs and sanitize tool outputs. Experiments on several benchmarks (AgentDojo, ASB, InjecAgent, τ-Bench) show near-zero attack success rates. The paper also points out flaws in existing benchmarks and suggests improvements.

**Strengths:**

The paper is clearly written and easy to follow.

The idea is simple and practical — the defense can be easily applied without retraining.

The benchmark analysis is detailed, and the authors identify real issues in existing evaluation setups.

**Weaknesses:**

The core idea (“minimize & sanitize”) is too simple and incremental, offering little novelty beyond existing “firewall” or “guardrail” defenses.

Most results come from benchmarks that the authors themselves criticize as flawed, so the findings feel self-contradictory and less convincing.

The paper lacks deeper insight or analysis about why the approach works and how it generalizes.

The work doesn’t propose new attack strategies or theoretical understanding — it’s mainly an engineering evaluation rather than a research contribution.

Overall, the contribution feels minor; it’s a straightforward application rather than a new idea.

**Questions:**

see weakness

---

> ### Author Response · Authors · 2025-11-20
> **Response to Reviewer MDkm (1/4)**
>
> We thank the reviewer for recognizing the simplicity and practicality of the baseline defense we evaluated (no retraining required, generalizes across benchmarks). We also thank the reviewer for appreciating the detailed analysis of benchmarks.
>
> >**W1:** The core idea is too simple and incremental. The work doesn’t propose new attack strategies or theoretical understanding — it’s mainly an engineering evaluation rather than a research contribution. The contribution feels minor; it’s a straightforward application rather than a new idea.
>
> **A1:** We agree that LLM-as-a-judge has been studied in jailbreaking and prompt injection contexts. However, nowhere in the prior work did we find concrete numbers for the firewall baseline on the standard tool-calling security benchmarks (AgentDojo [1], InjecAgent [2], ASB [3]). Results don't necessarily transfer from the Jailbreaking to Indirect Prompt Injection setting: for instance, DoomArena [4] showed that Llamaguard was unable to flag any of the indirect prompt injection attacks they proposed.
>
> **What is a Research Contribution?** Research contributions can take many forms, and are not limited solely to “novel attack strategies” or "novel defense methods". Contributions can also be made through thoughtfully examining existing assumptions in the field (e.g., complex defenses are needed to beat the benchmarks), carefully examining progress thus far (e.g., benchmark limitations), and proposing new directions and considerations that may be needed in the community based on empirical evidence (e.g., more challenging benchmarks). Most importantly, we believe that before proposing new methods, it is equally, if not more important, to extensively and carefully assess the current landscape. Meaningful progress can only be made with a clear understanding of the existing limitations.
>
> **Our Research Contribution is Answering a Scientific Question:** Our work is titled "Are Firewalls All You Need, or Stronger Benchmarks", and it is exactly the scientific question that we try to answer, through extensive empirical analysis on four popular security benchmarks. Our results jointly highlight that (i) a simple firewall-based defense baseline is enough to saturate current security benchmarks, (ii) current benchmarks have several flaws that we fix and provide best practices for future development, and (iii) stronger, more adaptive benchmarks are needed to meaningfully measure progress in agentic security.
>
> **Novel Findings:**
> - **Simple firewalls can easily saturate current benchmarks:** Despite being lightweight, the firewall baseline matches or exceeds the performance of more complex approaches across all 4 major agentic security benchmarks.
> - **The firewalls generalize across models:** We evaluate using four different LLM backbones (GPT-4o, LLaMA 3.3-70B, Qwen3-8B, Qwen3-32B), demonstrating that the firewall-based defense’s effectiveness does not hinge on a specific model or implementation trick.
> - **Current benchmarks have several flaws:** We find and fix the flaws and provide best practices for future development.
>     - For instance, because security benchmarks such as ASB [3] were previously published at ICLR 2025, reviewers refer to it as a strong benchmark for IPI attacks. However, our investigation uncovered several weaknesses and flaws in it (e.g. forcefully injecting an attacker tool in the agent’s planning, lack of arguments), which lead to misconstrued performance and hinder the meaningful measurement of future security progress.
>     - Similarly, the absence of any utility assessment in InjecAgent [2] (accepted at ACL 2024) limits its practical relevance, since an agent that fails to perform its intended tasks could trivially appear perfectly secure.
>     - We also uncover and fix implementation bugs in AgentDojo [1], identify missing or misleading metrics–findings that are new and important for the community.
>
> **Novel Recommendations and avenues for future research:**
> - We suggest that the community use our revised version of AgentDojo instead of the original one (due to minor flaws in evaluation metrics).
> - We ask AI security reviewers to no longer require InjecAgent and ASB numbers, because they may lead to misconstrued performance and hinder the meaningful measurement of future security progress.
> - We urge the community to always report Sanitizer scores as a simple but effective baseline when proposing a new defense method.
> - We urge the community to develop stronger, more adaptive benchmarks, because they are needed to meaningfully measure progress in agentic security.
> - The fact that obfuscation-based strategies can bypass even frontier LLM firewalls (GPT-4o) should be investigated in future work.
>
> **These are all new findings and recommendations that have not been previously published or discussed in the context of tool-calling agent security.**

---

> ### Author Response · Authors · 2025-11-20
> **Response to Reviewer MDkm (2/4)**
>
> >**W2:** Results on using benchmarks that authors themselves criticized are flawed, so the findings feel self-contradictory and less convincing.
>
> **A2:** As we state in the conclusion, the results on the revised Agentdojo are the ones that the community should really care about. We provide detailed results on both the original and revised AgentDojo benchmark (see Tables 7-9 and 10-12). On both revised and original versions of the benchmark, our firewalls achieve near-perfect security (Table 1 and Table 8) and outperform more sophisticated defenses such as CaMeL [5] and MELON [6].
>
> Despite minor issues, AgentDojo remains the strongest available benchmark. Our modifications there are limited to improving the utility evaluation (yielding 10% improvement in utility under attack), not changing the underlying tasks or attacks.
>
> On top of AgentDojo, we also evaluated our defense on InjectAgent and ASB, despite their flaws, because these 3 are currently the most widely adopted agentic security benchmarks in the literature. They have effectively become the standard for evaluating defenses, and, as in the case of MELON, reviewers routinely request results on these benchmarks [7].
>
> The popularity of these benchmarks makes our fixes even more crucial for ensuring consistency and reliability of reported results. **Hence, we believe that publishing at a venue such as ICLR would allow the community to benefit from these insights in a timely manner.**

---

> ### Author Response · Authors · 2025-11-20
> **Response to Reviewer MDkm (3/4)**
>
> >**W3:** The paper lacks deeper insight or analysis about why the approach works and how it generalizes.
>
> We appreciate the suggestion. We performed 3 additional quantitative analyses and 2 additional qualitative analyses:
>
> 1. **Latency analysis:** We present the time required by different LLM backbones to complete the Agentdojo evaluation.
>
> | Model              | Defense    | No attack (s) | With attack (s) |
> |--------------------|-----------|---------------|-----------------|
> | gpt-4o-2024-08-06  | none      | 510.74        | 5917.86         |
> | gpt-4o-2024-08-06  | minimizer | 1257.10       | 13277.98        |
> | gpt-4o-2024-08-06  | firewall  | 1697.19       | 20811.03        |
> | gpt-4o-2024-08-06  | combined  | 2384.43       | 25406.55        |
> | Qwen3-32B          | none      | 1280.41       | 11330.01        |
> | Qwen3-32B          | minimizer | 1421.15       | 11501.99        |
> | Qwen3-32B          | firewall  | 2194.99       | 46781.95        |
> | Qwen3-32B          | combined  | 2628.93       | 49893.98        |
>
> Further, we compare in detail with CaMeL since it is a state-of-the-art system-level defense that employs policies to ensure safety on 2 suites of Agentdojo (banking and Slack) using gpt-4o-2024-08-06. CaMeL requires a total of 8,417.25 seconds to solve the tool-knowledge attack runs, whereas our Firewall needs only 3,347.93 seconds. This corresponds to a 2.5x speed-up over CaMeL under the same attack setting.
>
> | Defense    | No attack (mean) | No attack (total) | Tool-knowledge (mean) | Tool-knowledge (total) |
> |-----------|------------------|-------------------|------------------------|-------------------------|
> | CaMeL     | 32.84            | 838.70            | 65.04                  | 8417.25                 |
> | Minimizer | 28.74            | 736.11            | 41.95                  | 4998.47                 |
> | Firewall  | **27.38**        | **699.90**        | **27.67**              | **3347.93**             |
> | Combined  | 46.75            | 1194.82           | 46.38                  | 5640.89                 |
>
> We have added this analysis in Tables 22 and 23.
>
>
> 2. **Cost overhead:** To evaluate the efficiency of different defenses, we measure the number of input and output tokens used per task on AgentDojo, using gpt-4o-2024-08-06 as the backbone model. We tokenize all messages using OpenAI’s tiktoken tokenizer [8]. Since the native tool-calling API does not expose how it tokenizes tool documentation, we exclude those tokens for all methods, previous ones, and ours. We do include all tokens from the firewall system prompts, ensuring their overhead is fully accounted for.
>
> We find that CaMeL incurs the highest token cost among all defenses, requiring 2.82x more input tokens and 2.73× more output tokens than the baseline (no defense) for the median task. In contrast, the Sanitizer alone requires only 0.51x the input tokens and 2.67x the output tokens, showing that it achieves similar or better security with far less prompt overhead, especially in terms of input token usage, which contributes significantly to latency and cost. Full statistics are shown in the table below.
>
> | Defense        | Type   | Mean tokens | Median tokens | Std. dev. |
> |----------------|--------|-------------|---------------|-----------|
> | None           | Input  | 3764        | 2004          | 4826      |
> | None           | Output | 303         | 271           | 174       |
> | Spotlighting   | Input  | 3914        | 2053          | 4698      |
> | Spotlighting   | Output | 295         | 268           | 169       |
> | Tool Filter    | Input  | 16562       | 14277         | 12804     |
> | Tool Filter    | Output | 313         | 293           | 152       |
> | Repeat Prompt  | Input  | 5854        | 2491          | 8176      |
> | Repeat Prompt  | Output | 387         | 301           | 324       |
> | CaMeL          | Input  | 13463       | 3643          | 21276     |
> | CaMeL          | Output | 1843        | 585           | 2830      |
> | **Sanitizer**      | Input  | 2036        | 1030          | 2618      |
> | **Sanitizer**      | Output | 1675        | 723           | 2557      |
>
> As shown in the above table, our Sanitizer is much more token-efficient than the system-level defense CaMeL. For input tokens, Sanitizer reduces the mean from 13,463 to 2,036 (≈ 6.6x fewer tokens) and the median from 3,643 to 1,030 (≈ 3.5x reduction). Output token usage is of the same order as CaMeL (mean 1,843 vs. 1,675), so overall, the Sanitizer achieves strong security with substantially lower prompt overhead. We have added this analysis in Table 24.

---

> ### Author Response · Authors · 2025-11-20
> **Response to Reviewer MDkm (4/4)**
>
> *(continuation of **A3**)*
>
> 3. **False Positive and Negative Analysis:** We provide the misclassification statistics that are important for assessing firewall reliability. On AgentDojo, we explicitly compute TP/FP/FN/TN for the Sanitizer:
>
> | Component      | Recall  | FNR   | FPR   | Precision | Accuracy |
> |----------------|---------|-------|-------|-----------|----------|
> | Sanitizer      | 99.53%  | 0.47% | 0.00% | 100.00%   | 99.82%   |
>
> As requested, we provide detailed misclassification statistics in the above table. The Sanitizer demonstrates extremely reliable behavior, with zero false positives and near-perfect recall. The Sanitizer shows a very small false negative rate (0.47%), meaning it misses only a handful of attacks, while maintaining 100% precision (i.e., never misclassifying benign responses). These results support our claim that the Tool Output Firewall is both effective and precise. We have added this to Table 25 in the paper.
>
> 4. **Failure examples:** We also provide examples where the Minimizer (Appendix G.5.1) and the Sanitizer (Appendix G.5.2) fail and cause a drop in benign utility. The main reasons are:
> (a) the Minimizer redacts arguments that are necessary to complete the user’s initial task, and
> (b) the Sanitizer incorrectly flags tool outputs as unsafe due to the presence of essential PII (e.g., an IBAN) required for solving the task.
>
> We think that we’ve already answered your questions in the weaknesses section. If not, please let us know and we will be happy to provide more explanations. In light of the analysis and experiments, we hope that you’ll consider raising your score.
>
> [1] Debenedetti, Edoardo, et al. "Agentdojo: A dynamic environment to evaluate prompt injection attacks and defenses for llm agents." Advances in Neural Information Processing Systems 37 (2024): 82895-82920.
>
> [2] Zhang, Hanrong, et al. "Agent security bench (asb): Formalizing and benchmarking attacks and defenses in llm-based agents." ICLR 2025.
>
> [3] Zhan, Qiusi, et al. "Injecagent: Benchmarking indirect prompt injections in tool-integrated large language model agents." ACL 2024.
>
> [4] Boisvert, Leo, et al. "Doomarena: A framework for testing ai agents against evolving security threats." arXiv preprint COLM 2025.
>
> [5] Debenedetti, Edoardo et al. “Defeating Prompt Injections by Design.” ArXiv abs/2503.18813 (2025).
>
> [6] Zhu, Kaijie et al. “MELON: Provable Defense Against Indirect Prompt Injection Attacks in AI Agents.” ICML (2025).
>
> [7] Melon Reviews ICML 2025. https://openreview.net/forum?id=gt1MmGaKdZ
>
> [8] OpenAI 2022. tiktoken: Fast BPE tokeniser for use with OpenAI’s models. url: https://github.com/openai/tiktoken.

---

> > ### Author Response · Authors · 2025-11-27
> > **Requesting rebuttal feedback**
> >
> > Dear Reviewer MDkm,
> >
> > Thank you again for your thoughtful review and constructive comments. As the discussion period will end in a few days, we wanted to kindly check whether our responses have addressed your concerns. We would be happy to clarify anything further if needed. We appreciate your feedback and look forward to hearing from you.
> >
> > If the clarifications resolve the issues you raised, we would be grateful for your updated assessment.

---

### Official Review · Reviewer_VeH2 · 2025-11-01

**Soundness:** 3
**Presentation:** 3
**Contribution:** 3
**Rating:** 6
**Confidence:** 4

**Summary:**

This paper presents a two-fold contribution. First, it proposes a simple, model-agnostic defense against indirect prompt injection (IPI) attacks called the "minimize & sanitize" firewall. This defense consists of a "Tool-Input Firewall" (Minimizer) to filter sensitive data from tool inputs and a "Tool-Output Firewall" (Sanitizer) to remove malicious instructions from tool responses. The authors demonstrate that this defense, particularly the Sanitizer, achieves state-of-the-art (SOTA) results, reducing the attack success rate (ASR) to ≈0 on AgentDojo and $\tau$-Bench and to ≲0.3 on InjecAgent. The paper's second, and arguably more significant, contribution is a rigorous critique of these same benchmarks. The authors reveal that the SOTA results are largely an illusion, as the benchmarks suffer from critical flaws, such as "forced attack-tool injection" in ASB and brittle utility metrics in AgentDojo. This makes them poor evaluators of true security. To prove their point, the authors develop a stronger, obfuscated (Braille-based) attack that successfully bypasses their own SOTA firewall, thereby highlighting the urgent need for stronger, more realistic security benchmarks.

**Strengths:**

1. The paper's primary strength is its dual contribution. It not only proposes a simple, effective, and model-agnostic defense (the Minimizer-Sanitizer firewalls) but also provides a rigorous critique of the very benchmarks used to measure success.
2. This paper uncovers flaws in ASB and AgentDojo that distort ASR and utility, and provide concrete fixes to make evaluations more trustworthy.
3. The proposed firewall defense is commendable for its simplicity and practicality. As a modular, model-agnostic approach, it serves as an excellent and easily replicable baseline.

**Weaknesses:**

1. Potential Data Contamination. While the reported results of the proposed defense are strong, this method relies primarily on frontier models (GPT-4o and Qwen3), and the paper does not analyze potential training–evaluation contamination (prior exposure to attack styles or benchmark artifacts). Could you replace the Minimizer/Sanitizer with an older model and report the performance? This would show whether the defense truly depends on frontier-model memorization.

Overall, this is an interesting and valuable paper that productively revisited progress on benchmarking prompt-injection attacks and defenses. It would be even better with a quantitative treatment of optimization-based adaptive attacks [1, 2]. Conceptually, these attacks should also serve as strong baselines, especially since many defenses in current benchmarks are largely static and plausibly vulnerable to adaptive optimization (e.g., tuning an adversarial suffix).

[1] Liu, Xiaogeng, et al. "Automatic and universal prompt injection attacks against large language models." arXiv preprint arXiv:2403.04957 (2024). \
[2] Pasquini, Dario, Martin Strohmeier, and Carmela Troncoso. "Neural exec: Learning (and learning from) execution triggers for prompt injection attacks." Proceedings of the 2024 Workshop on Artificial Intelligence and Security. 2024.

**Questions:**

For rebuttal, please refer to the weaknesses.

---

> ### Author Response · Authors · 2025-11-20
> **Response to Reviewer VeH2**
>
> We appreciate the reviewer for recognizing our work's dual contribution: (1) evaluating a strong baseline which is simple, effective, and model-agnostic defense and (2) providing a rigorous critique of the dominant agentic security benchmarks. We address their concerns below.
>
>
> >**Q1:** Potential Data Contamination in the latest models. Does the defense truly depend on frontier-model memorization?
>
> **A1:** We appreciate the reviewer’s concern regarding potential training–evaluation contamination due to our reliance on frontier models such as GPT-4o and Qwen3. However, we note that the knowledge cutoff for gpt-4o-2024-08-06 (we use this version of GPT-4o) is October 2023 [1], and for Llama 3.3, the knowledge cutoff date is December 2023 [2], while the benchmarks used in our evaluation were released after that date:
> AgentDojo: First commit February 29, 2024 [3]
> InjectAgent: Released February 19, 2024 [4]
> ASB: First commit October 2, 2024 [5]
>
> Thus, GPT-4o and Llama 3.3 70B could not have been trained on these benchmarks or attack formats. There is not data contamination.
>
> >**Q2:** Replacing the sanitizer with older models (GPT-4-turbo, Llama 3.3 70B)
>
> **A2:** To further strengthen our claim, we also performed an analysis where we changed the sanitizer to Llama 3.3 70B (cut-off knowledge date : December 2023) and GPT-4-turbo (cut-off knowledge date: December 2023). We present the evaluation on Agentdojo below.
>
> | Assistant Model        | Sanitizer Model          | BU (%)           | UA (%)           | ASR (%)          |
> |------------------------|--------------------------|------------------|------------------|------------------|
> | gpt-4o-2024-08-06      | gpt-4o-2024-08-06        | 67.68 ± 3.56     | 69.17 ± 2.47    | 0.02 ± 0.03      |
> | gpt-4o-2024-08-06      | Llama 3.3 70B            | 70.86 ± 8.46     | 62.20 ± 1.38     | 0.68 ± 0.03      |
> | gpt-4o-2024-08-06      | gpt-4-turbo-2024-04-09   | 69.32 ± 5.32     | 65.87 ± 0.89     | 0.62 ± 0.12      |
>
> We have added this table as an ablation to the PDF in Table 21 in Appendix G.1.
>
> We hope that this will help clarify that our method's effectiveness does not hinge on memorization or contamination, but rather on a generalizable prompting structure.
>
>
> >**Q3:** It would be even better with a quantitative treatment of optimization-based adaptive attacks.
>
> **A3:** We appreciate the suggestion. While optimization-based adaptive attacks can be useful for probing robustness under white-box assumptions, both attacks mentioned by the reviewer rely on gradient access. However, we intentionally frame our evaluation around a realistic deployment setting, where:
>
> The attacker does not have access to the internals of the agent or firewall (i.e., they are black-box),
> The agent may be closed-source or proprietary.
> No training set or gradients from the assistant agent are available.
>
> In such real-world scenarios, optimization-based attacks are often impractical, especially in multi-component agentic pipelines where end-to-end differentiability is unavailable. Our goal is to design a practical defense that is robust under these realistic threat models.
>
> That said, we agree that analyzing our method under optimization-based, white-box attacks would be a valuable direction for future work, especially for probing its theoretical limits.
>
>
> If our responses have addressed your concerns, we kindly ask you to consider revising your score. If not, we would greatly appreciate any suggestions on how we can further clarify or improve our work.
>
> [1] GPT-4o-2024-08-06 knowledge cut-off date. https://platform.openai.com/docs/models/gpt-4o
>
> [2] Llama 3.3 knowledge cut-off date. https://huggingface.co/meta-llama/Llama-3.3-70B-Instruct
>
> [3] AgentDojo first commit. https://github.com/ethz-spylab/agentdojo/commits/main/
>
> [4] ASB first commit. https://github.com/agiresearch/ASB/commits/main/
>
> [5] InjecAgent release. ​https://github.com/uiuc-kang-lab/InjecAgent/commits/main/

---

> > ### Author Response · Authors · 2025-11-27
> > **Requesting rebuttal feedback**
> >
> > Dear Reviewer VeH2,
> >
> > Thank you again for your thoughtful review and constructive comments. As the discussion period will end in a few days, we wanted to kindly check whether our responses have addressed your concerns. We would be happy to clarify anything further if needed. We appreciate your feedback and look forward to hearing from you.
> >
> > If the clarifications resolve the issues you raised, we would be grateful for your updated assessment.

---

> > > ### Comment · Reviewer_VeH2 · 2025-11-27
> > >
> > > Dear Authors,
> > >
> > > Thank you for your rebuttal. It addresses my concerns. Well done.

---

### Official Review · Reviewer_qyiU · 2025-11-01

**Soundness:** 2
**Presentation:** 3
**Contribution:** 2
**Rating:** 4
**Confidence:** 5

**Summary:**

This paper claims to provide a simple, effective, modular and model-agnostic defense for tool-calling agents based on: a Tool-Input Firewall (Minimizer) and a Tool-Output Firewall (Sanitizer). They demonstrate that their approach achieves 0% or the lowest attack success rate on four public benchmarks while maintaining high task success. They also found and fixed key flaws in widely popular benchmarks to enable more reliable evaluation in the agent attack community. Finally, they also provided case study about the failure of their own method to call for more stronger defenses.

**Strengths:**

1. The contribution of fixing existing benchmarks is very useful for future benchmarking in this field.

2. The method seems to work well on the 4 benchmarks, making them strong candidates for agent security defense.

3. this paper is easy to understand, and the demonstration is very good and illustrative.

**Weaknesses:**

1. The method is, despite its fancy names, imo, a pre-processor and a post-filter, which is not new. And I did not see any necessity to associate it with firewalls as inspiration.

2. The two filters (pre&post) seem only been built by a very short system prompt. thus, it's questionable why those system prompts serves for the claim of 'firewall is equipped with a robust system prompt'. How do you justify this? why that system prompt is robust? how did you choose those system prompt?

3. In 7 DISCUSSION, I understand stronger attacks can unsuperisingly bypass your firewalls. But how about other baselines? does the same attack also succeed on other baselines or it a flaw of your own methods? more discussion on this would be good.

**Questions:**

1. How about finetuning the two filters? Why is the system prompting enough?

2. What's the difference if the backbone model is not gpt4o? Any trade-off analysis?

---

> ### Author Response · Authors · 2025-11-20
> **Response to Reviewer qyiU (1/2)**
>
> We thank the reviewer for their positive comments on our paper’s writing and visualizations' clarity, comprehensive benchmarking, and appreciating our contributions towards fixing existing benchmarks.
>
>
> > **Q1:** Method is a pre-processor and a post-filter, no necessity to associate it with firewalls as inspirations
>
> **A1:** We borrow the term firewall from previous works [1] to emphasize the security role of our components rather than just their position in the pipeline. In network security, a firewall monitors and controls incoming and outgoing traffic according to predefined security policies, acting as a barrier between trusted and untrusted entities [CISA, 2023]. Our Tool-Input Firewall (Minimizer) plays an analogous role by stripping unnecessary or sensitive information from tool arguments before the tool is called, while the Tool-Output Firewall (Sanitizer) filters tool responses to remove potentially malicious or injected content before they are passed back to the agent. Together, they implement a “minimize and sanitize” barrier at the agent–tool boundary, separating trusted user intent from untrusted tool inputs and outputs. We provide a clarification about this at L49-50.
>
> We would like to emphasize that our goal in this work is not primarily to introduce a completely new defense, but rather to demonstrate that a simple LLM-based firewall is, in fact, a very strong candidate for agentic defense. Across four widely adopted security benchmarks, firewall baselines saturates and matches or outperforms many more sophisticated defenses. We believe this establishes a stronger baseline for agentic security, which has not been identified in prior work in this setting, to the best of our knowledge.
>
> However, our analysis also reveals numerous shortcomings in the very benchmarks the community has adopted to measure progress. This ranges from minor implementation bugs (e.g. it was impossible even for an oracle to achieve perfect utility in AgentDojo), to missing crucially important metrics (utility in InjecAgent), or unrealistic assumptions (e.g. hard-coded attacker tool calls in ASB).
>
> Collectively, we believe that these findings (a seemingly strong, simple baseline and fixing current benchmarks) provide novel and insightful results for the agentic security community.
>
>
> > **Q2:** The two filters (pre and post) appear to rely only on very short system prompts. This raises doubts about the claim that the firewall is equipped with a robust system prompt. How do you justify this robustness?
>
> **A2:** We thank the reviewer for pointing this out. We simply meant that the same prompts achieve high security and utility over all 4 benchmarks. We agree that describing the system prompt as “robust” is misleading, and we will remove this wording from the manuscript. We have revised the sentence at L182 to: “This firewall is equipped with a system prompt and is also provided with the original trusted user task, the tool name, and its input arguments.”
>
>
> > **Q3:** How were these prompts chosen?
>
> **A3:** The prompts were manually crafted. Our goal was not to heavily optimize them for any specific dataset, but to keep them short, transparent, and easily applicable across all four benchmarks. We therefore focused on simple, high-level instructions that (i) clearly specify the firewall’s objective (minimize sensitive information / sanitize unsafe content) and (ii) remain agnostic to any particular benchmark or task template. The exact prompts for both firewalls used for evaluation are given in Appendix B.
>
>
>
> [1] Abdelnabi, Sahar et al. “Firewalls to Secure Dynamic LLM Agentic Networks.” ArXiv abs/2502.01822 (2025).

---

> ### Author Response · Authors · 2025-11-20
> **Response to Reviewer qyiU (2/2)**
>
> > **Q4:** A stronger attack can unexpectedly bypass the firewall. Does it pass through other defenses as well, or is it a flaw of the firewall?
>
> **A4:** It is a flaw of the firewall. In this (user task 0, injection task 0) of the travel suite, the sanitizer detects the suspicious braille-encoded instructions, but instead of removing them, it decodes them to plain language, which then hijacks the agent.
> We ran the Braille indirect prompt injection against all baselines. The attack only succeeds with the GPT-4o-2024-06-06 firewall, and fails with every other defense, as well as with no defense:
> | Defense       | Success |
> |---------------|-------|
> | None          | 0     |
> | Repeat Prompt       | 0     |
> | Spotlighting  | 0     |
> | Tool filter   | 0     |
> | Sanitizer      | 1     |
>
> Importantly, this aligns with our paper’s positioning: we are not proposing a new defense method, but analyzing an important class of baselines (firewalls) and showing that they can introduce new, defense-specific failure modes that current security benchmarks do not capture. This supports the need for stronger security evaluations for indirect prompt injections.
>
> > **Q5:** How about finetuning the two filters? Why is the system prompting enough?
>
> **A5:** We chose not to fine-tune the Tool-Input and Tool-Output Firewalls to reflect a realistic security deployment scenario, where we may not have access to a representative training set or prior knowledge of the attack formats. In many real-world agentic deployments, unseen or evolving prompt injection strategies make it difficult to rely on static fine-tuned models.
>
> Instead, we use manually crafted prompts (shown in Appendix B) that are deliberately simple and generalizable across benchmarks. Importantly, the firewall is not limited to a system prompt alone; both a system prompt and a user prompt are provided to the LLM. This lightweight prompting setup enables strong performance without dataset-specific tuning, making it easy to deploy in settings with minimal prior assumptions about the attacks.
>
> We believe this highlights the practical strength of our method: despite its simplicity and lack of fine-tuning, it achieves strong results across all benchmarks, suggesting that simple, well-designed prompting can go a long way in enforcing agent–tool boundary security.
>
> >**Q6:** What's the difference if the backbone model is not gpt4o? Any trade-off analysis?
>
> **A6:** We explicitly study this by varying the backbone beyond GPT-4o. In particular, we report results for Llama 3.3-70B, Qwen3-8B, and Qwen3-32B on all four benchmarks: original AgentDojo in Tables 10 (utility without attack), 11 (ASR), and 12 (utility under attack), original ASB in Tables 13 (no-attack utility) and 14 (utility and ASR under attack), InjectAgent in Table 19, and Tau-Bench in Table 20. There was not enough space in the main paper, so we suggest the reviewer to check out the appendix for the relevant tables.
>
> Across these backbones, we observe the same qualitative trade-off: stronger models achieve higher absolute utility, but in all cases, our firewalls substantially reduce ASR while maintaining high benign and under attack utility. Importantly, the prompts and setup are not tuned per model, yet the defense remains effective, suggesting that our approach is not specific to GPT-4o. We clarify this in the main text by adding a short summary of these trends and explicit cross-references to Tables 10–20 in L344-348.
>
> If we have addressed all of your concerns, we hope that you’ll consider raising your score. If not, we hope that you can give us some guidance on how we could better address your concerns.

---

> > ### Author Response · Authors · 2025-11-27
> > **Requesting rebuttal feedback**
> >
> > Dear Reviewer qyiU,
> >
> > Thank you again for your thoughtful review and constructive comments. As the discussion period will end in a few days, we wanted to kindly check whether our responses have addressed your concerns. We would be happy to clarify anything further if needed. We appreciate your feedback and look forward to hearing from you.
> >
> > If the clarifications resolve the issues you raised, we would be grateful for your updated assessment.

---

### Author Response · Authors · 2025-12-03
**Summary of discussion**

Dear reviewers and AC,

We thank the reviewers for their thoughtful and constructive feedback. We are encouraged that multiple reviewers found our firewall defense to be **simple** (VeH2, MDkm, eTDx), **effective** (VeH2, eTDx), and **model-agnostic** (VeH2), highlighting its **practicality** (VeH2, MDkm) and **ease of application without requiring retraining** (MDkm), as demonstrated through **comprehensive evaluations** (qyiU, eTDx). The benchmark analysis was recognized as **detailed and rigorous** (VeH2), and we appreciate the **acknowledgment of our concrete fixes to existing flawed evaluations** (qyiU, VeH2, MDkm). Reviewers also noted the paper is **clearly written, easy to follow, and supported by illustrative examples** (qyiU, MDkm). Finally, we appreciate the **recognition of our dual contribution**: introducing a strong baseline defense while also strengthening the agent-security evaluation through fixes to existing benchmarks (VeH2).

We responded to all comments individually. Below, we summarize the updates and clarifications.

### Summary of main changes:

- **Data-contamination and backbone concerns (VeH2):** We added a discussion of model knowledge cutoffs vs. benchmark release dates and new experiments where the Sanitizer is replaced with Llama 3.3-70B and GPT-4-turbo. These results are similar to GPT-4o, supporting that our defense does not rely on benchmark memorization by frontier models. *(Appendix G.1)*
-  **Practical deployment analysis (eTDx, MDkm):** We added latency measurements and token-cost analysis for all defenses. These analyses highlight that the Sanitizer provides the best security while having substantially lower overhead in terms of tokens as well as lower latency than system-level defenses. *(Appendix G.2 and G.3)*
- **Misclassification statistics (eTDx, MDkm):** We added detailed metric evaluation, including false positive, false negative, precision and accuracy for the Sanitizer on AgentDojo. Our results demonstrate that the tool-output firewall is both effective and precise. *(Appendix G.4)*
- **Impact on benign utility (eTDx):** We provided a qualitative analysis of tasks where benign utility drops while using the Sanitizer. Our analysis highlights that most BU loss stems from agent variability, not over-aggressive filtering by the Sanitizer. *(Table A)*

### Additional Clarifications:
- **Does Braille attack bypass all defenses (qyiU):** We have added an analysis showing that the Braille-based attack presented in the paper specifically reflects a failure of the Sanitizer, not other defenses. This clarifies the class of obfuscated attacks capable of bypassing our firewall and more rigorously characterizes its failure modes.
- **Provided results on both original and revised benchmark versions on multiple LLMs:** In response to the comment by reviewers (qyiU, MDkm), we clarify that the **paper reports results on both the original benchmark implementations and our corrected versions**, evaluated across **multiple models** (Qwen3 32b, Qwen3 8b, Llama3.3 70b, GPT-4o) to ensure fairness and reproducibility, while also highlighting the importance of our proposed fixes.
- **Rationale for avoiding finetuning (qyiU):** We avoid fine-tuning to mirror realistic security settings where training data and attack formats are unknown. Evolving injection strategies can quickly invalidate static fine-tuned models. Instead, we rely on simple, general prompts (system + user), which require no dataset-specific tuning yet perform strongly across benchmarks, highlighting the practicality of prompt-based firewalls for agent–tool security.

### Novelty Clarification

We emphasize that the novelty of our contribution lies in answering the core scientific question posed in our title: “Are Firewalls All You Need, or Stronger Benchmarks?” Through extensive evaluation of a lightweight firewall baseline across four agent-security benchmarks and multiple LLM backbones, we show that it effectively saturates current benchmarks. More importantly, we identify and address several flaws in benchmarks widely adopted by the community, and we provide best-practice recommendations and stronger evaluation protocols for future research in agentic security.

We again thank all reviewers (qyiU, VeH2, MDkm, eTDx) for their careful reading and constructive suggestions. We hope these changes and additions address the main concerns, and we remain happy to clarify any remaining concerns.

---

### Meta-Review · Area_Chair_iJtA · 2026-01-03

**Summary:**

This paper studies indirect prompt injection (IPI) attacks in tool-using LLM agents and asks a clear scientific question: Are simple firewall-style defenses sufficient, or are stronger benchmarks needed to meaningfully evaluate agent security? It uses the simple firewall-like defense method and show it can outperform other methods on existing benchmark. It also strengthens the agent-security evaluation through fixes to existing benchmarks.

Reviewers provide 4,6,2,2 and have concerns such as experimental details, novelty, data contamination, performance against stronger attacks and so on. After rebuttal, there are still some concerns regarding to the novelty and contribution of this paper. Specifically, reviewer eTDx pointed out some inaccurate argument proposed by authors ( e.g., nowhere in the prior work did we find concrete numerical data for the firewall baseline on the standard tool-calling security benchmarks.) and still concerns about novelty.

Overall, the main concerns are 1. little novelty and 2. deeper insight or analysis about why the approach works and how it generalizes.

The AC partially disagrees with some of the reviewers’ characterization of the work. In particular, the AC does not believe that the paper has little contribution. AC still thinks this paper indeed has some contributions to the domain as described by the authors. However,  despite these contributions, the overall impact is not yet sufficient for acceptance at ICLR.  The contribution would need to go further—either by substantially strengthening the benchmarks (e.g., through stronger or adaptive attackers that better approximate worst-case real-world threats), or by providing a more general evaluation framework or more robust defense mechanisms. In its current form, the paper offers important insights and corrections, but these contributions alone are not yet large or forward-looking enough to meet the bar for ICLR acceptance. The AC believes that extending this work along the directions above would significantly increase its impact and value to the community.

**Reviewer Concerns:**

Reviewer qyiU's concerns

1. Fancy name.
Authors clearly explained it.
2. Question about two filters, prompt selection.
Authors have addressed them by providing a clear explanation.
3. Stronger attack.
Authors conducted experiments to show the flaw of the sanitizer.

Review VeH2's concern:
1. Data contamination.
Authors conducted new experiments to clearly address it.


Review MDkm's concern:
1. little novelty.
Authors clearly explain the contribution of this paper.
2. findings are self-contradictory and less convincing.
Authors have explains it well. This concern is addressed.
3. Lacks deeper insight or analysis about why the approach works and how it generalizes.
Authors have analyzed the latency, overhead, false positive rate and failure examples. But authors did not explicitly answer why this approach works and how it generalizes.
4.  contribution is minor.
Same as 1.

Review eTDX's concern:
1. limited novelty.
Authors clearly explain the contribution of this paper.
2. Degradation in benign utility
Authors provided analysis. This concern is addressed.
3. Analysis on practical deployment considerations:
Authors conducted new experiments. This concern is addressed.

Overall, the main concerns are 1. novelty and 2. deeper insight or analysis about why the approach works and how it generalizes.

The AC partially disagrees with some of the reviewers’ characterization of the work. In particular, the AC does not believe that the paper has little contribution. AC still thinks this paper indeed has some contributions to the domain as described by the authors. However, despite these contributions, the overall impact is not yet sufficient for acceptance at ICLR.  The contribution would need to go further—either by substantially strengthening the benchmarks (e.g., through stronger or adaptive attackers that better approximate worst-case real-world threats), or by providing a more general evaluation framework or more robust defense mechanisms. In its current form, the paper offers important insights and corrections, but these contributions alone are not yet large or forward-looking enough to meet the bar for ICLR acceptance. The AC believes that extending this work along the directions above would significantly increase its impact and value to the community.

**Reviewer Scores:**

VeH2 (6): Confirmed concerns addressed → remains positive.
qyiU (4): Likely remains borderline/weak accept after clarifications.
MDkm (2):  May increase point to 4 due to novelty expectations and analysis on why this method works.
eTDx (2):   May increase point to 4 due to the novelty.

---

### Decision · Program_Chairs · 2026-01-26

Reject